# Vibro-Acoustic Signatures of Various Insects in Stored Products

**DOI:** 10.3390/s24206736

**Published:** 2024-10-19

**Authors:** Daniel Kadyrov, Alexander Sutin, Nikolay Sedunov, Alexander Sedunov, Hady Salloum

**Affiliations:** 1Department of Civil, Environmental & Ocean Engineering, Stevens Institute of Technology, Hoboken, NJ 07030, USA; asutin@stevens.edu (A.S.); nsedunov@ieee.org (N.S.);; 2Department of Civil Engineering, University of Houston, Houston, TX 77204, USA; hsalloum@uh.edu

**Keywords:** acoustic, insect, detection, stored product, rice, grain

## Abstract

Stored products, such as grains and processed foods, are susceptible to infestation by various insects. The early detection of insects in the supply chain is crucial, as introducing invasive pests to new environments may cause disproportionate harm. The STAR Center at Stevens Institute of Technology developed the Acoustic Stored Product Insect Detection System (A-SPIDS) to detect pests in stored products. The system, which comprises a sound-insulated container for product samples with a built-in internal array of piezoelectric sensors and additional electret microphones to record outside noise, was used to conduct numerous measurements of the vibroacoustic signatures of various insects, including the *Callosobruchus maculatus*, *Tribolium confusum*, and *Tenebrio molitor*, in different materials. A normalization method was implemented using the ambient noise of the sensors as a reference, to accommodate for the proprietary, non-calibrated sensors and allowing to set relative detection thresholds for unknown sensitivities. The normalized envelope of the filtered signals was used to characterize and compare the insect signals by estimating the Normalized Signal Pulse Amplitude (NSPA) and the Normalized Spectral Energy Level (NSEL). These parameters characterize the insect detection Signal Noise Ratio (SNR) for pulse-based detection (NSPA) and averaged energy-based detection (NSEL). These metrics provided an initial step towards the design of a reliable detection algorithm. In the conducted tests NSPA was significantly larger than NSEL. The NSPA reached 70 dB for *T. molitor* in corn flakes. The insect signals were lower in flour where the averaged NSPA and NSEL values were around 40 dB and 11 dB to 16 dB, respectively.

## 1. Introduction

Stored products refer to a variety of non-perishable agricultural commodities kept in storage for future use, including grains, legumes, nuts, and spices. They are essential for food security, and nutrition, and also represent significant economic value. These products are also vulnerable to infestation by various insect pests, which can cause significant losses in quantity and quality, threaten health and safety, as well as pose the risk of introducing invasive species to new environments. Effective means of inspecting products for pests are crucial throughout the global supply chain to ensure the integrity of the product and mitigate greater infestation.

### 1.1. Stored Products and Insect Pests

The main types of grains are cereals, pulses, and oilseeds. Common cereals include wheat, rice, corn, rye, millet, and barley [1]. Pulses, such as beans, lentils, peas, and chickpeas, are the edible seeds of the *Leguminoseae* family and are commonly consumed by humans and domestic animals in dried form. Legumes that are not considered pulses are used for edible oils such as peanuts, soybeans, and alfalfa [2]. Other oilseeds include sunflower seeds, rapeseeds, and sesame seeds [3].

The target commodities have a high risk of infestation by various insects. The types of stored product pests are grouped based on their eating behaviors. Internal feeders have larvae that feed entirely within the kernels of whole grain and palletized products. They can remain visually undetected until adults emerge from the kernels, leaving burying holes into a hollow kernel and powder residue. The internal feeders include the rice weevil, *Sitophilus oryzae*, granary weevil, *Sitophilus granaries*, Angoumois grain moth, *Sitotroga cerealella*, and the lesser grain borer, *Rhyzopertha dominica*. External feeders feed on the outside of the grain, chewing through the outer seed coat and then devouring the inside. They include the drugstore beetle, *Stegobium paniceum*, cigarette beetle, *Lasioderma serricorne*, and the khapra beetle, *Trogoderma granarium*. Scavengers feed on the product only after the seed has been broken, either mechanically or by some other insect. Examples include the *Tribolium confusum*, *Tribolium confusum*, red flour beetle, *Tribolium castaneum*, sawtoothed grain beetle, *Oryzaephilus surinamensis*, and the Mediterranean flour moth, *Ephestia kuehniella*. Secondary pests, such as mealworms, *Tenebrio molitor*, feed on materials that are deteriorating, damp, or have mold growth present [4].

Stored product pests (SPP) can infest crops at various stages, including during growth in the field, transportation, and storage. An infestation can cause quantitative loss from direct grain weight and volume depletion. Qualitative loss can occur from excrement, webbing, and dead bodies contaminating the product. Insects can consume the grain embryos, which decreases the protein content of the grains. They can also affect the chemical components of the grains, promoting infection of the skin and body of the kernels, attracting further pathogenic microorganisms, and lowering seed viability. The presence of insects can also damage storage containers, facilities, and equipment [5]. High temperatures promote the infestation rate and insects produce moisture and heat while living in the product which further improves the growth conditions [6].

The Food and Agriculture Organization of the United Nations (FAO) estimates that 20% of global food production is lost to herbivorous insects [7]. Other sources state that insects cause up to 9% and 20% in postharvest losses in developed and developing countries, respectively [8,9]. Crop losses were estimated to cost 470 billion USD per year in 2001 [10]. Introducing alien insects into new environments can result in more significant ramifications as the species becomes invasive without natural predators. Invasive species cost the U.S. 2 trillion USD from 1960 to 2020, costing 1 billion USD a year from 2010 to 2020 in damages, losses, and prevention management [11]. Chemically treating the food product can be harmful to humans as well as the environment and is not always effective as insects can develop resistance to natural or artificial toxins. Inspecting products at critical points of the supply chain is instrumental to prevent the spread of pests and ensure product quality.

### 1.2. Inspection Standards for Pests in Stored Products

The International Organization of Standards (ISO) guides the means of storing cereals and pulses as well as controlling and detecting attacks by pests. The ISO 6639 standard [12] specifies that a container is deemed infested if an insect is found within 1 kg of the material sample. During inspection, the collected material sample is divided into 200 g to 300 g test samples. If insects are not found, the material is considered to be free of insects. The USDA determines an infestation if a 1 kg sample contains two or more live insects [13].

### 1.3. Inspection Methods for Stored Product Pest

The ISO 6639 standard describes several methods for detecting insects in stored grain products, focusing on carbon dioxide analysis, the ninhydrin method, the whole grain floatation method, as well as acoustic and X-ray methods [12]. Other methods can be categorized based on the type of technology they leverage. The application of these inspection methods has to balance the accuracy of detecting and classifying pests of various life stages with safeguarding product quality, minimizing associated costs, ensuring the efficiency of supply chain movement, and limiting the amount of damage or loss during examination.

#### 1.3.1. Physical Methods

The current main method for detection is through manual visual inspection, where a chosen commodity is emptied on an inspection table and a trained inspector insects insect body parts, fragments, metabolites, and signs of eggs, discoloration, and webbing networks on the grain surface. This method is often aided with physical tools such as sieves, tweezers, magnifying glasses, and intense lights [13].

Probe traps can be placed at different grain depths for extensive amounts of time and are periodically removed to determine the number, species, and density of the infestation. This method is restricted to easily accessible locations within the container, and limits the temporal availability of results, and the capture rate is influenced by the insect species, the type of grain, as well as the temperature and humidity [14]. These traps include the Berlese–Tullgren trap and the Tamil Nadu Agricultural University probe traps [15,16].

Visual lures and pheromones are often used to improve the performance of traps. Visual lures could be lights, such as incandescent, fluorescent, and ultraviolet, that attract insects from dark or dimly lit surroundings [17]. Chemical attractants, natural and synthetic, are used to mimic the aggregation and sex pheromones secreted by insects [14]. The Electronic Grain Probe Insect Counter (EGPIC), rolling sieve Insectomat 5K and the electrical conductance method speed up and automate the detection process but are expensive and perform poorly with infested kernels, insect eggs, and young larvae [18,19,20,21,22]. The floatation method, although effective, is destructive and requires a large sample size, which is not always available [23].

#### 1.3.2. Chemical Methods

Carbon dioxide evolution monitors changes in the ${\mathrm{CO}}_{2}$ levels within a space emitted by the metabolic actions of insects inside or outside kernels. Although it is effective for measuring multiple grain bins simultaneously as well as monitoring bulk grain storage, it is time-consuming and ineffective for moisture contents higher than 15% [24,25].

Uric acid is the main element in insect excrement and can be used as a tracing element in the storage of food grains to determine an infestation. The uric acid levels can be determined through paper chromatography, fluorometric methods, colorimetric, gas-liquid chromatography (GLC), thin layer chromatography (TLC), high-performance liquid chromatography (HPLC), and enzymatic methods [14].

The electric nose (E-nose) consists of an array of gas sensors with varying sensitivities, a data acquisition unit, and pattern recognition software. The sensors detect the volatile compounds and react to the changing electrical properties [26]. Although not developed for insect detection, the E-nose was able to assess the storage age and the insect infestation of the lesser grain borer, *Rhyzopertha dominica*, in wheat [27].

#### 1.3.3. Spectral Imaging

Non-destructing X-ray imaging with systems such as the TrapsSesotec RAYCON X-ray system and the X-ray Inspection System AD-4991 can visualize invisible insects within stored food grains as well as fruits and vegetables as well as provide damaged and underdeveloped products [28,29]. Systems feature an X-ray source, converter, imaging system, and an isolated casing. Near-infrared reflectance (NIR) can determine the concentration of constituents such as water, protein, fat, and carbohydrates based on the absorption differences of electromagnetic wavelengths and can provide the ability to detect live and dead insects at various life stages as well as classify non-infested and infested kernels but require calibration for every new testing batch [14,30,31]. Thermal imaging visualizes the infrared radiation energy, or heat signature, emitted by the object. The produced image, a thermogram, can detect damaged grains, foreign materials, as well as internal and external infestation by quantifying the temperature differences [17]. Deep learning algorithms further improve the accuracy of these methods [5].

#### 1.3.4. Microwave Methods

Experiments described in [32] demonstrated an application of a system originally developed for the detection of a hidden termite infestation based on microwave Doppler radar operating at 24 GHz. The movement of the insect is detected by radiating a microwave signal and observing the change in the frequency of the received reflection, which increases as the insects move slightly closer or decreases as they move further away from the sensor. The system was able to detect the cigarette beetle, *Lasioderma serricorne*, the sawtoothed grain beetle, *Oryzaephilus surinamensis*, the black carpet beetle, *Attagenus unicolor*, and the red flour beetle, *Tribolium castaneum* within a corn meal and flour mixture.

#### 1.3.5. Acoustic Methods

Acoustic detection methods rely on the sound produced by living internal and external insects during movement, mating, and mastication. Microphones, piezoelectric sensors, laser vibrometers, micromechanical sensors (MEMS), accelerometers, ultrasonic transducers, and seismic sensors are often used with amplification and filtering algorithms. The sound generation and capture depend on the properties of the insect, such as size, feeding habits, speed, and activity; the properties of the material, specifically density, kernel size, and hardness, as well as environmental factors, including humidity, temperature, and background noise. Acoustic sensors can monitor for hidden infestations without destroying the product and have been applied to detect insects in other hidden substrates, including soils, trees, interior structures of plants, and underground. They can also provide real-time and non-destructive information on the insect population density, species composition, and spatial distribution [33].

Acoustic signals are characterized by their intensity, duration, and spectral characteristics. Measurements from the presence of insects often take the form of impulses, which are short-duration, wide-frequency band signals. The attributes of the pulses include the spectral content, duration, and amplitude. Bursts are a series of pulses that can be enumerated by their number and rate within a time period.

Spectral and temporal features can characterize the bioacoustics of specific species and their life stages. The Khapra beetle, *Trogoderma granarium*, which can cause from 5% to 15% and up to 70% in extreme cases, was most dominant in the frequency bands from 1000 Hz to 8000 Hz in wheat grains. The adult and larvae occupied different bands, 3500 Hz to 4500 Hz and 1000 Hz to 1700 Hz, respectively. Female adults were dominant in the 1000 Hz to 3000 Hz and 4000 Hz to 5000 Hz frequency bands [34]. The peak of energies for an adult and larva of the rice weevil, *Sitophilus oryzae*, in wheat grain were in 1800 Hz to 3000 Hz and 1300 Hz to 2000 Hz, respectively [35].

The impulse rate, the number of impulses per unit of time, has effectively been used to identify an infestation, distinguish species and life stages, and overcome the influence of noise. The burst could range from 50 to 200 impulses [36] and last from 3 ms to 30 ms [37]. Adult lesser grain borers, *Rhyzopertha dominica*, and red flour beetles, *Tribolium castaneum*, produced 37 and 80 times more sound than their larvae [37]. Larger amounts of impulses could be used as an indication of higher infestation numbers [38]. Background noise tends to be continuous or with isolated pulses, which can be filtered out by setting a threshold for the number of impulses per unit time [33].

Machine learning (ML) algorithms have been applied to improve the accuracy of acoustic insect detection. Although raw acoustic data could be fed directly into the training algorithms, often the data are preprocessed to extract features such as Mel-frequency cepstral coefficients (MFCC), spectral centroid, spectral flatness, spectral roll-off, linear predictive cepstral coefficients (LPCC), and line spectral frequencies (LSF). The data can also be converted into spectrograms, two-dimensional graphical representations of the magnitude of the signal at frequencies over time, which can be used with traditional image classification models [39]. ML requires large datasets to train and test models. The Animal Sound Archive [40], InsectSet32 [41], InsectSet42 and InsectSet66 [42], and the Singing Insects of North America [43] all feature recordings of insects; however, they are not species commonly found infesting store products. The BugByte sound library is the only publicly available dataset of insect sounds with recordings of stored product pests in various materials [44].

The Acoustic Location Fixing Insect Detector (ALFID) system is vertically mounted, and grains are gravity-loaded and unloaded. Sixteen piezoelectric acoustic sensors are linearly oriented to identify the number of feeding larvae and their location within the sample. Signals from the sensors are amplified by 80 dB and bandpass filtered from 1000 Hz to 10,000 Hz. Amplitude threshold detection identifies the arrival time of the signal, and the location is determined by a detection order algorithm. The system was tested with up to three larvae of the rice weevil, *Sitophilus orzae*, in 1 kg of wheat and was able to detect the larvae with 64% accuracy and 8% false positives [45]. The ALFID system was patented in 1996 [46].

The Beetle Sound Tube was designed for semi-permanent monitoring of silos, flat stores, and big bags of grain for early, low levels of infestation and to provide remote reports to keepers. The system is comprised of a series of perforated metal tubes that are installed into the grain mass. The tubes feature an insect trap that could be inspected for insect classification, an acoustic sensor, and a climate sensor to measure changes in temperature and humidity. The signal processing algorithm utilized $\frac{1}{3}$ octave analysis, calculating the energy level for 16 $\frac{1}{3}$-octave bands from 250 Hz to 8000 Hz. If the energy level exceeded a threshold of 25 dB within 250 Hz to 800 Hz, 20 dB within 800 Hz to 2500 Hz, or 15 dB within 2500 Hz to 8000 Hz, above the background noise, it was determined to be a sign of insect presence. The sensor and algorithm successfully detected the sawtoothed grain beetle, *Oryzaephilus surinamensis*, within barley in flat storage. Through cross-correlation, the sensor could also determine the level of infestation [6].

The Acoustic Emission Consulting AED 2010L is a portable, battery-operated system with piezoelectric sensor mounted on a metal probe that is inserted into a grain sample. It was able to detect one to two insects per kg of wheat grain with 72% to 100% accuracy and were able to predict the population densities of the tested live specimens including the rice weevil, *Sitophilus oryzae*, the lesser grain borer, *Rhyzopertha dominica*, the confused flour beetle *Tribolium confusum*, the sawtoothed grain beetle, *Oryzaephilus surinamensis*, the rusty grain beetle, *Cryptolestes ferrugineus*, the khapra beetle, *Trogoderma granarium*, and the cigarette beetle, *Lasioderma serricorne*, through chi-squared tests and machine learning classifiers [47,48]. This system was available for 2000 USD but is no longer in production [49].

The Postharvest Insect Detection System (PDS) consisted of electret microphones embedded at the center of a plexiglass base onto which the grain sample is placed. The system was tested with the rice weevil, *Sitophilus oryzae*, and the red flour beetle, *Tribolium castaneum* [49]. The Computerized Acoustical Larval Detection system, patented in 1988, features a bowl design that holds 1 L of grain sample with a diaphragm at the bottom connected through a conduit that couples it to a microphone [50]. The Ultrasonic Insect Detector, also termed the Purdue Insect Feeding Monitor, was patented in 1990 [51]. The active sensor was able to sense the feeding habits of the cowpea beetle, *Callosobruchus maculatus*, inside cowpea seeds at the 40 kHz range. Since sound waves attenuate at high frequencies when traveling through the grain, the sensor is only effective at distances less than a few millimeters [52]. It was also able to monitor the development of the rice weevil, *Sitophilus oryzae*, in maize kernels [53].

One of the challenges of using acoustic sensors for detecting insects in stored products is the interference of external noise from the surrounding environment, such as factory machinery, vehicles, wind, and other sources. External noise can mask the low-intensity sounds produced by insect feeding and movement, which typically range from 20 dB to 80 dB SPL (sound pressure level) at frequencies below 10 kHz. Acoustic detectors were shielded using a multi-layered enclosure that attenuates sound by 70 dB to 85 dB between 1 kHz to 10 kHz. This method enables reliable detection of internally feeding larvae in grain samples at inspection facilities with high noise backgrounds [54].

### 1.4. Previous Work by the STAR Center

The Sensor, Technology, and Applied Research (STAR) Center Laboratory at Stevens Institute of Technology developed several iterations of acoustic sensors and algorithms to detect insects while inspecting stored products during transportation and import. They tested the sensors on live insects including the larger cabinet beetle, *Trogoderma inclusum*, mealworm and darkling beetle, *Tenebrio molitor*, the corn earworm, *Helicoverpa zea*, *Copitarsia* larvae, and the khapra beetle, *Trogoderma granarium*, within a variety of materials including grains, vegetables, and herbs.

The first acoustic system comprised a test container outfitted with ceramic piezoelectric sensors that output four separate audio channels into an external preamplifier and an analog-to-digital converter (ADC) before reaching a computer for recording and post-processing. The test container is located within a large box with two plastic containers separated by sound-absorbing foam in between. The design provided from 50 dB to 80 dB of sound attenuation for frequencies above 500 Hz.

After determining that the acoustic signals recorded from insect presence were concentrated in the frequency band from 400 Hz to 1000 Hz, the group developed a processing chain algorithm to detect the presence of insects in the recorded signals. The signal is split into two bands, 200 Hz to 600 Hz and 600 Hz to 1500 Hz, for the noise and insect signal, respectively. The envelopes of the signals are calculated using a Hilbert transform. The difference between the two envelopes is used as a detection metric. A detection is determined when the metric exceeds a threshold value. This system was able to detect a *Trogoderma inclusum* larva and adult with 100% and 70% accuracy, respectively, and reported a 0% false positive rate when testing without live insects within 1 kg of rice [55].

Several years later, the group redesigned the apparatus to be more portable and integrated the preamplifier and data acquisition electronics into the container. A large Pelican case was modified with acoustic dampening material to provide 60 dB of external noise dampening for frequencies above 500 Hz. The 40 cm by 60 cm testing enclosure within featured three measurement channels, each utilizing two piezoelectric modules. The updated detection algorithm involves applying a bandpass filter from 500 Hz to 5000 Hz, smoothing the squared signals through a moving time average filter with a time window of 50 ms, and normalizing to the average intensity. A detection was determined when the signal exceeded the background noise level with a 5 dB threshold [56].

### 1.5. Summary and Need for an Improved System

The inspection methods and devices described above effectively detect insects in stored products but have various setbacks and limitations, especially in usage at ports of entry and exit. Although the manual visual inspection method is the most common and has the most human oversight, it is labor intensive, time-consuming, and requires a substantial workforce to service large volumes of product. Probes and traps, even when assisted with visual lures, pheromones, and automation, require a generous amount of time to ascertain the infestation level and are limited to the locations where they are placed. Although they are efficacious for silos and long-term storage, they are not suitable for quick examinations. The flotation method destroys the product. The other methods and devices, including the sampling and sieving machines, electrical conductance, chemical, and spectral methods, are expensive and require specialized equipment and training.

Acoustic sensors are non-destructive and can provide information on insect population density, species composition, and spatial distribution. The devices available vary in design based on their usage in storehouses and inspection sites. Although external noise can interfere with efficacy, efforts are made to mitigate the impact through soundproofing and signal processing.

## 2. Materials and Methods

### 2.1. A-SPIDS Sensor Description

The Acoustic Stored Product Insect Detection System (A-SPIDS) improves on the previous work by the STAR Center. It aims to provide a low-cost, portable, fast, and non-destructive method for stored product inspection at port facilities. This new iteration has a smaller footprint and weight. The electronics are incorporated into the body of the sensor, are battery-powered and wirelessly rechargeable, and transmit data over Wi-Fi to a computer for processing and storage. The system features physical insulation and also uses external microphones to reduce the impact of external noise.

The system is made up of three parts: a main body, an acoustically treated lid, and a base that provides vibration isolation and wirelessly charges the battery. The total system weighs 9 kg and is the size of a large shoebox with a length, height, and width of 30 cm, 24 cm, and 19 cm, respectively. Figure 1 shows an annotated diagram of the system:

The main body consists of a sample testing container inside a 3D-printed enclosure. The internal container is 26 cm long, 15 cm wide, and 3.5 cm tall and can hold up to 1.5 L of product. The container size is large enough to satisfy inspection sampling requirements and stay in range for optimal insect detection. Previous sensors were able to detect the rice weevil, *Sitophilus oryzae*, and an adult red flour beetle, *Tribolium castaneum*, up to 10 cm to 15 cm and 18.5 cm away from the grain, respectively [57,58]. Since insects tend to be more active on the surface of the substrate [59], the maximum distance from the piezoelectric discs to any portion of the sample is set within this range. Insulation lines the periphery of the sensor to physically reduce the impact of external noise. The insulation is made of 1.875 cm thick acoustic foam and molded silicone.

During usage, the testing sample is placed into the receptacle, and the acoustically treated lid is tightly closed. The bed of the testing container is lined with 32 piezoelectric discs grouped into eight modules that capture the sounds and vibrations of the insects moving, masticating, and mating within the product. The discs are 27 mm in diameter, 0.33 mm thickness, and made of piezoelectric ceramic lead zirconate titanate and brass [60]. These modules receive the signal on channels zero through three, corresponding to their respective location within the container. Traditional acoustic microphones, which transfer the received signals through channels four to seven, are placed throughout the body of the system to capture and mitigate external noise and vibration from affecting the results. The microphones are POM-2735P-R with a diameter of 6 mm, a height of 4.7 mm, and a sensitivity of −35±2 dB [61]. Figure 2 shows the piezoelectric layout:

The bottom of the main body houses the electronics for data acquisition, communication, and power. The output from the piezoelectric and microphones is sent through a custom amplifier and then converted to an I2S stream using TI PCM1808 analog-to-digital converter (ADC) made by Texas Instruments based Dallas, TX, United States and sourced from DigiKey [62]. The data is collected by a Raspberry Pi 3 A+ single-board computer sourced from Adafruit and manufactured at the SONY UK Technology Centre, United Kingdom [63] using a miniDSP MHCStreamer, a digital USB sound card manufactured in Hong Kong and sourced from the manufacturer that converts the data received over I2S into a digital stream [64]. The sampling rate was set to 24 kHz. The Raspberry Pi is connected to a Wi-Fi access point and transmits data to the acquisition computer that accesses the same network through a Mikrotik mAP model RBmAP2nD router manufactured in Latvia and sourced from Amazon [65]. The system is powered by a battery that can maintain the system for several hours and charges through a receiver coil when placed on the platform base. The electronics are organized on an EMI shielding plate, safeguarding the amplifiers and ADCs from interference. Figure 3 shows a diagram of the electronics:

The platform base of the system accommodates the transmitting coil that wirelessly charges the main system battery. It is powered by a 24 V power supply. The sensor can be lifted off and operated without the charging platform.

The intended use of the acoustic sensor is within a variety of environments, including storage facilities, warehouses, and transportation. These locations are often noisy due to machinery, vehicles, and people. Although noise sound pressure levels (SPL) in an office might be from 35 dBA to 50 dBA [37], factory noises can reach 110 dBA [66].

The A-SPIDS sensor is designed to mitigate the influence of external noise through soundproofing and leveraging the acoustic microphones in the detection algorithm. In the prototype model, the microphones were placed in different areas to compare the efficacy of the insulation. Microphones four and five are collocated to the piezoelectric modules within the internal storage container, channel six is placed inside the body, and channel seven is located on the body of the sensor.

The sensor was tested in varying levels of natural and artificial noise. Opening and closing the sensor, placing the sample into the container, settling the material kernels, and the migration of the insect during testing all produced significant disturbance and were trimmed from the recordings. Ambient noise was prevalent from laboratory activity, HVAC units, flying aircraft, vehicular traffic, and landscaping equipment. Pre-recorded simulated noise, including sweeps, white noise, loud conversations, crowds, and factory noise [67,68,69] was played through a speaker, and the noise level was measured using an Extech 407730 Digital Sound Level Meter manufactured and sourced from Teledyne FLIR (Wilsonville, OR, USA) based in placed near the sensor. The simulated noise recordings were increased in steps from 60 dBA to 100 dBA in 10 dBA steps. These tests were conducted with and without live insect specimens to examine the efficacy of the sound insulation as well as the influence of external noise on the detection, false positive, and false negative probabilities detailed in the future publications.

Figure 4 shows the SNR of the acoustic microphone channel five, colocated with the piezoelectric sensors within the testing container juxtaposed with the microphone channel seven placed on the external body of the sensor, while exposed to 80 dBA white noise.

Although the internal microphones were still able to capture external noise, especially below 2000 Hz, comparing the internally placed microphones, channel four and five, to the externally placed microphone, channel seven, showed that the insulation was effective in suppressing the noise by almost 30 dB. This level of sound suppression is within the limits achieved in other studies but with significantly lower material thicknesses [54].

### 2.2. Tested Insects and Materials

The target species, especially invasive and determined to be quarantine species, are difficult to obtain for laboratory testing. Several commercially available and easily attainable insects were chosen and purchased based on their similarity to the sizes, qualities, and behaviors of pest species targeted during inspections.

The cowpea weevil or cowpea seed beetle, *Collosobruchus maculatus*, is a member of the leaf beetle family, *Chrysomelidae*, and a significant pest of stored legume seeds such as beans, cowpeas, and mung beans. The adult is a small, round beetle, 2 mm to 4 mm in length, brownish with black, pale spots and stripes. Adult females lay eggs on the surface of the bean, and the larvae dig into the seed, feeding on the cotyledons during their development. Although adults have a lifespan of 12 days, the colony continues to reproduce until the depletion of the storage [7]. Flying versions of the species can disperse to new locations, including directly to cowpea fields. Economic losses caused from *Collosobruchus m.* are estimated to be 35%, 7% to 13%, and 7% in Central America, South America, and Kenya, respectively [70]. The weevil was responsible for up to 24% losses in stored pulses in Nigeria [71]. The burrowing behavior of the cowpea beetle mimics the behavior of other internal feeders, including the lesser grain borer, *Rhyzopertha dominica*, and the rice weevil, *Sitophilus oryzae*.

The confused flour beetles, *Tribolium confusum*, are arthropods commonly found infesting farinaceous material such as rice and flour. They range from 2 mm to 4 mm in length and are reddish-brown with club-like antenna endings. A secondary pest, the larvae and adult forms attack damaged grain, causing a grey tint, producing odor, and encouraging mold growth. The eggs are white, microscopic, and challenging to detect among the flour they inhabit. The lifecycle from egg to adult takes 40 to 90 days, and adults can live up to three years [72].

The *Tenebrio molitor*, a species of the darkling beetle, is known in its larval form as the mealworm. The larvae are typically 1 cm to 2 cm in length, have a dark brown head, and a segmented body with a hard exoskeleton. The larvae are yellow-brown and are often used as fishing bait. The adult beetle is 1.25 cm to 1.8 cm in length and is a shiny brown or black color. The mealworm is native to Europe, Asia, and North America [73]. Females can lay up to 600 eggs during their lifetime. Darkling beetles secrete benzoquinones from their abdominal glands, resulting in unpleasant odors. A secondary pest, the *T. molitor* larva eat the germs of the grain but can also feed on a wide variety of products, including tobacco and mold [7]. Table 1 describes the details of the tested insects:

During testing, 1 L of clean material was transferred into a thin plastic bag and placed into the sensor container. After ensuring that the grains had settled and recording several minutes of silence, a single insect was carefully relocated into the material. This setup matched the minimum conditions set by ISO and USDA standards for determining an infested material and surpassed previous studies which utilized from five up to fifty insects in similar volumes [45,49]. Several minutes of insect activity were recorded for each test set. This provided enough time to avoid capturing noise from independently moving kernels and increased the likelihood of capturing insect activity [37,45,49].

Common typically inspected stored products were chosen to represent the range of densities and kernel sizes. The materials included *Avena sativa* in the form of oatmeal, short-grain white rice, *Oryza sativa*, *Triticum aestivum* in the form of wheat groats and flour, and *Zea mays* in the form of cornmeal and Corn Flakes cereal. Table 2 describes the recording time per insect and material:

The insect–material pairings are not fully balanced as a result of varying life stages and availability of the insects, the need to test in clean and uncontaminated materials, and the time and budget constraints of the project.

## 3. Results

### 3.1. Acoustic Signatures of Insects in Different Materials

The data presented in Table 2 were trimmed to records with 1 min duration and the analysis below is based on samples with minimal background noise. Spectrograms, the visualization of the frequency content over time, were generated with a window size of 1024 samples and are shown in the 0 Hz to 8000 Hz frequency band. Figure 5 shows the spectrograms for high, medium, and low strength signal examples.

Figure 5a shows the spectrograms from all four piezoelectric sensor channels lining the bottom of the container during a recording of the *Tenebrio molitor* in rice. The relatively stronger signals in channel zero of Figure 5a indicate that the insect was located in the leftmost area of the system. The other channels show low signal due to longer propagation distance between the insect source and the respective sensor. Figure 5b shows the spectrogram of a medium strength signal generated by the *Callosobruchus maculatus* in oatmeal. Figure 5c shows the spectrogram of a lower strength signal made by the *Tribolium confusum* in flour. Only the spectrograms of the piezoelectric sensors with the highest signal strength are shown for the medium and low strength examples.

Spectra plots show a snapshot of the frequency content magnitude and were generated with a 1024 sample Blackman–Harris window averaged over the 60 s recording. Figure 6a displays the spectra of the four piezoelectric channels during the high-strength signal. Figure 6b shows the spectra of the high, medium, and low strength examples. Both figures compare the spectra to the spectra of reference noise, mainly produced by the sensor self-noise, when no insect is present.

In all the examples, the pulses propagating to the nearest sensor had the maximal amplitude and spectra relative to their counterparts which recorded weaker signals. The system lid was closed during recording so insect location could not be verified. Since the tested insects remained on the surface of the material, the minimum distance from the center surface of a piezoelectric sensor to the top of the material would be 5 cm.

The differences between insect signal spectra and the reference noise are presented in Figure 7 and can be interpreted as the signal-to-noise ratio (SNR) of the insect spectra:

It can be seen in the presented examples that the SNR is above zero in the frequency band from 500 Hz to 6000 Hz. This bandwidth was chosen for the signal analysis in the time domain during the statistical comparison of the insect signal amplitude and energy for the various insects and materials.

The Butterworth bandpass filter enables the extraction of the insect signal within this frequency band [74]. Equation (Equation 1) describes the magnitude response function of this filter:(1)$$M\left(\omega \right)=\frac{1}{\sqrt{1+{\left(\frac{\omega_{c}^{2}-\omega^{2}}{\omega \;\cdot \;\omega_{\Delta }}\right)}^{2n}}}$$
$$\omega_{c}=\sqrt{\omega_{1}\;\cdot \;\omega_{2}}\;\omega_{\Delta }=\omega_{2}-\omega_{1}$$
where $\omega_{c}$ is the center frequency, $\omega_{1}$ and $\omega_{2}$ are the lower and higher angular frequencies, respectively, and *n* is the order of the filter [75].

Figure 8 shows the waveform of the signal for *Callosobruchus maculatus* in oatmeal with an 500 Hz to 6000 Hz Butterworth 10th order bandpass filter applied.

### 3.2. Normalization of Signal Amplitude

Although the developed piezoelectric sensors are highly sensitive with relatively low noise levels, they are not calibrated and the recorded signals cannot be evaluated in the standard units for measurement such as μPa, dB
20reμPa, or for acceleration in vibration measurement $m/s^{2}$. The system self-noise, captured when no insects or background noise was present, was utilized as a reference level to compare the insect signal energy and amplitudes. This reference signal is shown in the spectra figures.

The envelope of the insect signal in the time domain was extracted by applying a Hilbert transform to the signal after imposing the Butterworth 10th order bandpass filter with a bandwidth from 500 Hz to 6000 Hz. The Hilbert transform, *H*, of the signal S(t) is defined in Equation (Equation 2) [76]:(2)$$\hat{S}\left(t\right)=H\left\{S\left(t\right)\right\}=\frac{1}{\pi }\int_{-\infty }^{\infty }\frac{S(\tau )}{t-\tau }d\tau$$
where *t* is the time and τ is the integration variable. Equation (Equation 3) shows the analytical signal, Z(t), obtained by combining the original signal with the Hilbert transform:(3)$$Z\left(t\right)=S\left(t\right)+i\hat{S}\left(t\right)$$

The envelope of the signal is then calculated by taking the magnitude of the analytical signal, Z(t), as shown in Equation (Equation 4):(4)$$S_{E}=\left|z\left(t\right)\right|=\sqrt{S^{2}+{\hat{S}}^{2}}$$

This procedure was conducted on the insect signals and the self-noise reference recordings for each channel. The root-mean-square (RMS) of the processed self-noise signal was utilized as the normalization factor and is defined in Equation (Equation 5):(5)$$\mathrm{RMS}\left(S\right)=\sqrt{\frac{1}{N}\sum_{i=1}^{N}S_{i}^{2}}$$
where $S_{i}$ is the *i*-th sample of the signal *S* and *N* is the number of samples. The normalization equation is presented in Equation (Equation 6):(6)$$S_{n}=\frac{S_{E,\mathrm{BPF}\;500-6000\;\mathrm{Hz}}}{\mathrm{RMS}(N_{E,\mathrm{BPF}\;500-6000\;\mathrm{Hz}})}$$
where the normalized signal $S_{n}$ is the envelope of the bandpass-filtered signal $S_{E,\mathrm{BPF}}$ divided by the root-mean-square of the enveloped bandpass-filtered reference self-noise $N_{E,\mathrm{BPF}}$. Figure 9 shows a diagram of the signal normalization process:

Visualizations of the normalization process for the high, medium, and low strength signals are presented in Figure 10.

### 3.3. Metrics for Signal Comparison

In order to compare the normalized insect signals across the different insects and materials, two metrics were derived utilizing the amplitude and spectral energy components of the respective signals.

The Normalized Strong Pulse Amplitude (NSPA) was calculated by taking the root-mean-square value of the normalized envelope within the 50% to 90% of the maximum and converting it to decibel amplitude. The channel with the highest NSPA was determined to be the one closest to the insect. Equation (Equation 7) describes the calculation of the NSPA:(7)$$\mathrm{NSPA}=20\log_{10}\left(\mathrm{RMS}\left(S_{c}\right)\right)\;\mathrm{where}\;0.5S_{c,\max}\leq S_{c}\leq 0.9S_{c,\max}$$
where $S_{c}$ is the envelope amplitude from channel *c*, RMS is the root-mean-square function, and $S_{\max}$ is the maximum envelope amplitude during the analyzed record.

Figure 10 shows the normalized envelopes of the insect signals in the different materials and highlights the analysis region for the NSPA calculation in red as well as distinguishes the calculated NSPA value.

The second parameter used for comparison is the Normalized Signal Energy Level (NSEL) in the frequency band 500 Hz to 6000 Hz. The total energy of the signal in the frequency band was calculated from the power spectral density (PSD), examples of which are presented in the Figure 6a. This procedure is equivalent to the calculation of the sound pressure level (SPL) in a definite frequency band [77]. Similarly to the amplitude normalization, the NSEL is then normalized to the sound pressure level of the reference noise level in the same frequency band. Equation (Equation 8) describes the calculation of the NSEL:(8)$$\mathrm{NSEL}={\mathrm{SPL}}_{500-6000\;\mathrm{Hz}}-{\mathrm{SPL}}_{\mathrm{noise},\;500-6000\;\mathrm{Hz}}$$

Figure 11 shows the distribution the NSPA and NSEL for the different insect and material pairings in a box plot. The box plot centers on the median value of the insect pulses in the respective material and the extents are the 25th and 75th percentiles. Table 3 details the averaged values of the metrics.

Figure 11 shows that in the majority of the tested commodities, the strongest signal amplitudes and energy were observed for the larger insects *Tenebrio molitor* and its larval form. Small insects produced much smaller signals. The exception to this relationship was within flour, where all tested insects generated similar relatively low signals. This can be attributed to the specifics of the small particle vibration under the actions of the moving insects. Visual observation of the insects moving in the flour showed that the larger insects had difficulty moving in the material, frequently coating themselves and getting stuck in the powder. A theoretical model describing the vibration and sound generation by insects in material was not found in any publication and might be a subject for future research.

### 3.4. Alternative Normalization Method

Although the self-noise reference signal was an effective way to normalize the insect signals, it requires a series of clean calibration recordings without insects or external noise. This is not always available, such as in the BugBytes dataset where the recordings do not feature a signal without insects to use as a reference signal for self-noise. An alternative method to determine self-noise was utilized by associating any signal below the median value of the signal amplitude as noise. The original signal was then normalized by dividing it by the RMS of the median of the assumed noise signal.
(9)$$S_{n}=\frac{S_{E,\mathrm{BPF}\;500-6000\;\mathrm{Hz}}}{\mathrm{RMS}(N_{s})}\;\mathrm{where}\;N_{s}<\mathrm{median}\left(S_{E,\mathrm{BPF}\;500-6000\;\mathrm{Hz}}\right)$$
Figure 12 compares the reference signal extraction using self-noise and the median reference method using the *Tenebrio molitor* in wheat groats.

The median reference extraction was effective at matching the self-noise signal, especially in the 500 Hz to 6000 Hz band demonstrating that this method can be used when a clean signal is not available.

### 3.5. Extending Analysis to the BugBytes Dataset

The developed approach can be applied stored product insect data collected by other sensors. The BugBytes dataset contains recordings of a single *Plodia interpunctella* larva in dry dog food and multiple *Sitophilus oryzae* larvae in wheat groats using a piezoelectric disc sensor. Although the recording system is not described in the dataset, previous publications by the authors suggest that the sensor is designed to record insects directly on the piezoelectric disc so the distance could not exceed 1 mm [78].

Since the piezoelectric disc sensor and system utilized for the BugBytes recordings are different from the A-SPIDS and the recordings did not feature a signal without insects to use as a reference signal for the self-noise, the median normalization method was applied to the recordings in the “Stored Product Insect movement and feeding sounds recorded for insect detection and monitoring studies” portion of the BugBytes dataset. The recordings include single larva and multiple larvae of the *Plodia interpunctella* in dry dog food, multiple larvae of the *Sitophilus oryzae* larvae in wheat groats, multiple *Sitophilus zeamais* adults and larvae in maize, and multiple *Prostephanus truncatus* adults and larvae in maize. The sensors featured accelerometers, piezoelectric discs, Polyvinylidene difluoride (PVDF) sensors, ultrasonic sensors, and microphones.

The *Sitophilus oryzae*, lesser grain borer or rice weevil larvae are internal feeders and develop while feeding within the grain. The larvae are 1 mm to 4 mm in length and are similar in size to the smaller insects recorded with the A-SPIDS [79]. Figure 13 shows a spectrogram of the *Sitophilus oryzae*, the rice weevil, larvae in wheat groats from the piezoelectric disc sensor:

Similarly to the recordings made by the A-SPIDS sensor, the BugBytes recording also features the impulses generated by the insect presence. These impulses are greatest in magnitude from 500 Hz to 8000 Hz and this frequency range was chosen for the median normalization method as well as NSPA and NSEL calculations. Figure 14 shows the spectra of the multiple *Sitophilus oryzae*, the rice weevil, larvae in wheat groats:

Figure 15 shows the normalized envelope of the *Sitophilus oryzae*, the rice weevil, larvae in wheat groats with the highlighted NSPA analysis region and calculated value. The NSPA measurement of the *Sitophilus oryzae* larvae in wheat groats was calculated to be 37.49 dB and the NSEL measurement was 19.43 dB. These values are comparable to the smaller insects in the same material from the A-SPIDS data.

The NSPA values were calculated for the other recordings in the BugBytes dataset and are presented in Table 4:

The results of the NSPA analysis demonstrated the effectivity of using the median normalization method to compare insects signals across different sensors and materials. The results from the piezoelectric sensors of the BugBytes dataset are consistent with the value ranges of the A-SPIDS sensor, but the A-SPIDS sensor was able to detect single at distances beyond 10 cm with a higher sensitivity. Although the adult stage of the *Sitophilus zeamais* had higher NSPA values than its larval stage, the opposite was observed for the *Prostephanus truncatus* adults and larvae. Recordings of the *Sitophilus oryzae* larvae in wheat groats across the different sensors revealed that the accelerometer had the highest sensitivity, followed by the PVDF film sensor, the piezoelectric sensor, and the ultrasonic sensors. Although the 30 kHz ultrasonic sensor was observed to have the lowest sensitivity, smaller than its 40 kHz counterpart, the insects could have been less active during the recording. The little amount of recorded data in the BugBytes dataset does not allow for greater statistical comparison of the sensors or further generalization of acoustic insect signatures, but the results are consistent with other publications [78].

## 4. Conclusions

The Acoustic Stored Product Insect Detection System (A-SPIDS) was developed to address a capability gap in a portable, fast, and non-destructive system to detect insect infestations in stored products such as grains, pulses, and seeds. Recordings using the sensor were conducted of live insect specimens including the cowpea beetle, *Callosobruchus maculatus*, the confused flour beetle, *Tribolium confusum*, and the larval mealworm and adult darkling beetle stages of the *Tenebrio molitor* in oatmeal, *Avena sativa*, rice, *Oryza sativa*, cornmeal and corn flakes, *Zea mays*, and wheat groats and flour, *Triticum aestivum*.

As the first step towards an optimal insect signal detection algorithm, the Normalized Strong Pulse Amplitude (NSPA) and the Normalized Signal Energy Level (NSEL) were introduced to use the normalization of the system self-noise to estimate the signal-to-noise ratio (SNR) values based on the pulse amplitude and spectra, respectively. Results from conducted tests showed that the NSPA is much higher than the NSEL and an amplitude-based insect detection method is more sensitive than a spectral-based alternative. Furthermore, lower signals were observed for insects in flour, demonstrating the level of difficulty in detecting pests in the material. This methodology was applied to the stored product insect recordings available in the BugBytes dataset, showcasing its validity to roughly estimate the detection performance of other sensors. The results of this work allowed for the development of an insect detection algorithm with external noise reduction capabilities that will be presented in following papers.

## Figures and Tables

**Figure 1 sensors-24-06736-f001:**
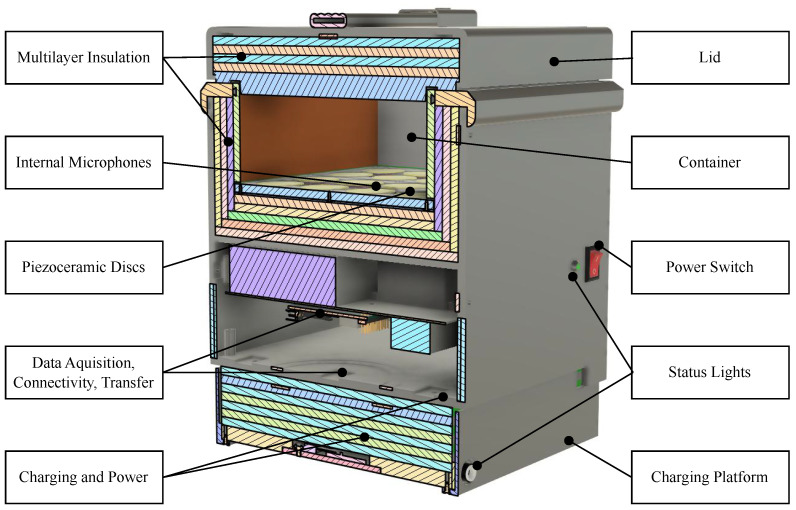
Acoustic Stored Product Insect Detection System with section view of internal container, sensors, and electronics.

**Figure 2 sensors-24-06736-f002:**
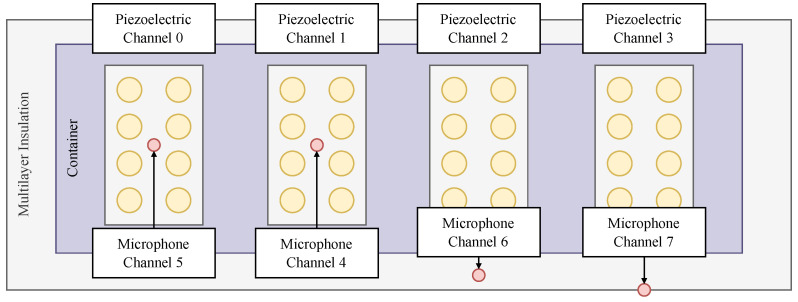
Diagram of the A-SPIDS system channel layout.

**Figure 3 sensors-24-06736-f003:**
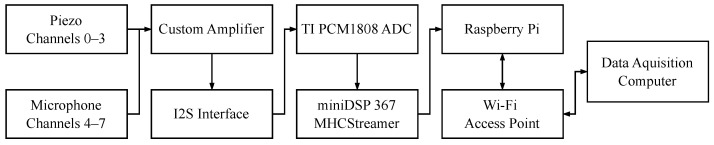
Diagram of system electronics.

**Figure 4 sensors-24-06736-f004:**
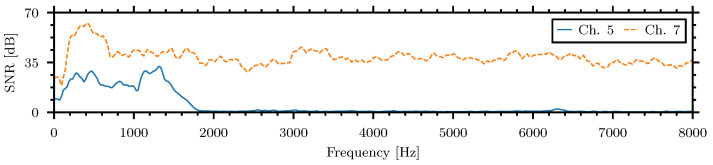
Spectra of external microphones while playing simulated external noise at 80 dBA.

**Figure 5 sensors-24-06736-f005:**
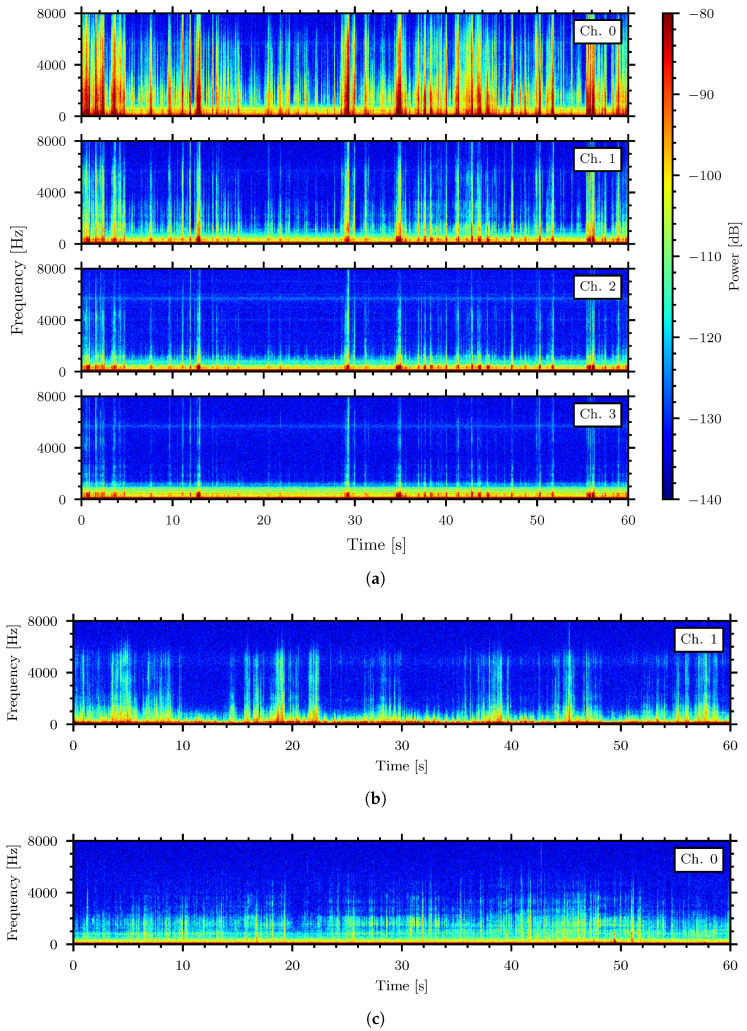
Spectrograms of high strength (**a**), medium strength (**b**), and low strength (**c**) signals. (**a**) High strength signal—*Tenebrio molitor* in rice, (**b**) Medium strength signal—*Callosobruchus maculatus* in oatmeal, (**c**) Low strength signal—*Tribolium confusum* in flour.

**Figure 6 sensors-24-06736-f006:**
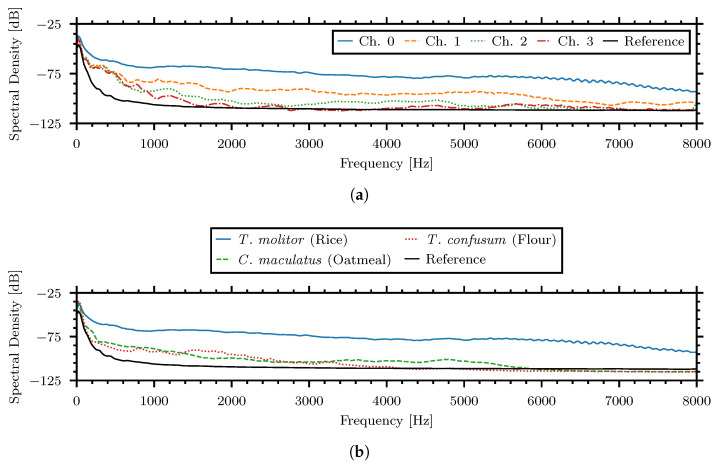
Spectra of the example insect–material pairings. (**a**) Spectra for four piezoelectric sensors of high strength signal—*Callosobruchus maculatus* in oatmeal, (**b**) Low strength signal—*Tribolium confusum* in flour.

**Figure 7 sensors-24-06736-f007:**
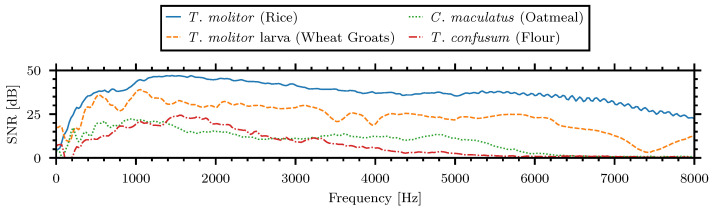
The differences between insect signal spectra and the reference noise, signal-to-noise ratio (SNR), of the example insect–material pairings.

**Figure 8 sensors-24-06736-f008:**
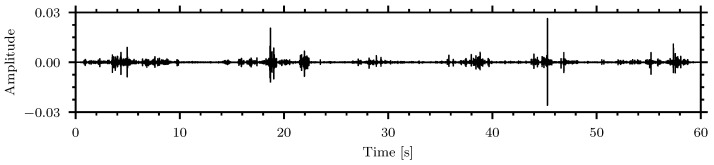
Filtered waveform of *Callosobruchus maculatus* in oatmeal.

**Figure 9 sensors-24-06736-f009:**
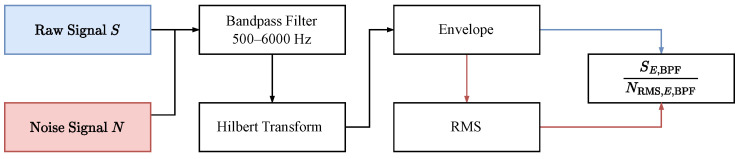
Diagram of the signal normalization process.

**Figure 10 sensors-24-06736-f010:**
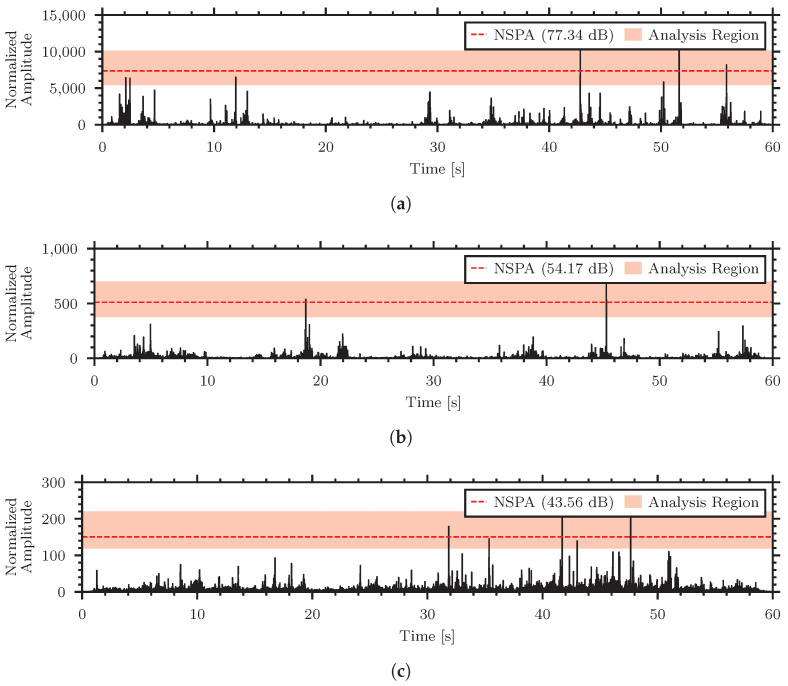
Normalized envelopes of the insect signals in different materials. (**a**) High strength signal—*Tenebrio molitor* in rice, (**b**) Medium strength signal—*Callosobruchus maculatus* in oatmeal, (**c**) Low strength signal—*Tribolium confusum* in flour.

**Figure 11 sensors-24-06736-f011:**
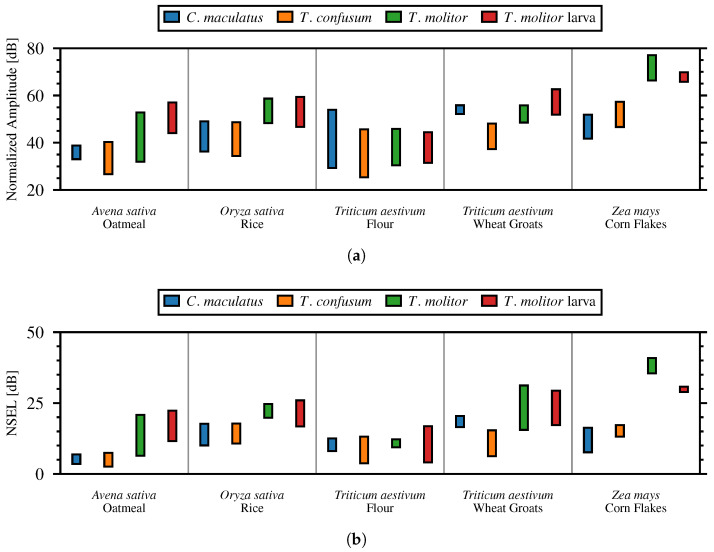
NSPA (**a**) and NSEL (**b**) for the different insect and material pairings. (**a**) Normalized Strong Pulse Amplitude (NSPA), (**b**) Normalized Signal Energy Level (NSEL).

**Figure 12 sensors-24-06736-f012:**
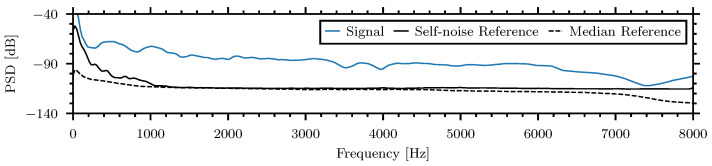
Spectra of *Tenebrio molitor* larvae in wheat groats comparing self-noise and median reference methods.

**Figure 13 sensors-24-06736-f013:**
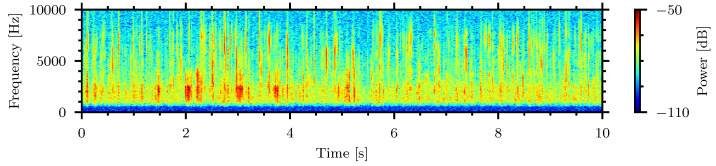
Spectrogram of *Sitophilus oryzae* larvae in wheat groats recorded by the piezoelectric sensor from the BugBytes dataset.

**Figure 14 sensors-24-06736-f014:**
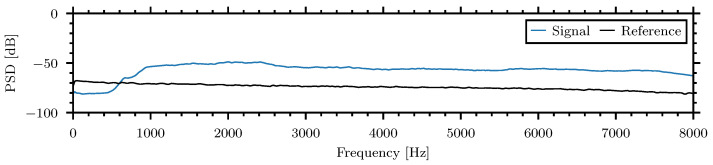
Spectra of *Sitophilus oryzae* larvae in wheat groats recorded by the piezoelectric sensor from the BugBytes dataset.

**Figure 15 sensors-24-06736-f015:**
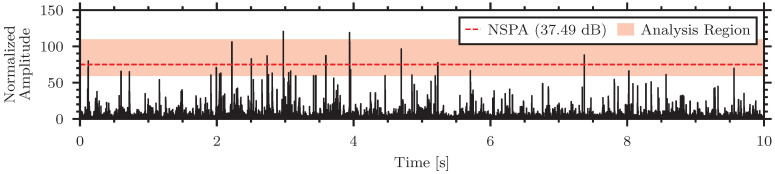
Normalized filtered envelope of *Sitophilus oryzae* larvae in wheat groats recorded by the piezoelectric sensor from the BugBytes dataset.

**Table 1 sensors-24-06736-t001:** Description of tested live specimens.

Scientific Name	Common Name	Size [mm]	Weight [mg]	Density [kg/L]
*C. maculatus*	Cowpea beetle	4×1×1	2.1	0.525
*T. confusum*	Confused flour beetle	4×2×2	1.0	0.625
*T. molitor* larva	Mealworm	20×2×2	130.0	1.625
*T. molitor*	Darkling beetle	15×5×3	160.0	0.711

**Table 2 sensors-24-06736-t002:** Duration of recordings in minutes for insect–material pairings with silence. The duration of recordings with external noise are accommodated with parenthesis.

	*Avena sativa*	*Oryza sativa*	*Triticum aestivum*		*Zea mays*
	**Oatmeal**	**White Rice**	**Wheat Groats**	**Flour**	**Corn Flakes**
*C. maculatus*	12	258 (12 noise)	13	20 (1 noise)	17
*T. confusum*	39	274 (16 noise)	189	55 (1 noise)	16
*T. molitor*	68	15	9	8	13
*T. molitor* larva	46 (4 noise)	268 (132 noise)	51 (2 noise)	128	43 (5 noise)
*No insect*	4	219 (64 noise)	142 (2 noise)	24	23

**Table 3 sensors-24-06736-t003:** Averaged values of NSPA and NSEL in dB.

		*Avena sativa*	*Oryza sativa*	*Triticum aestivum*		*Zea mays*
		**Oatmeal**	**Rice**	**Flour**	**Wheat Groats**	**Corn Flakes**
NSPA	*C. maculatus*	39.7	46.2	42.6	58.4	49.3
	*T. confusum*	35.4	45.9	39.5	43.9	53.9
	*T. molitor*	45.6	62.7	42.5	60.1	70.3
	*T. molitor* larva	56.8	56.8	39.6	61.0	70.5
NSEL	*C. maculatus*	9.9	17.3	12.4	24.2	14.4
	*T. confusum*	7.4	18.2	12.8	13.0	17.3
	*T. molitor*	16.5	32.8	15.5	28.3	38.4
	*T. molitor* larva	23.1	25.9	11.2	29.5	34.8

**Table 4 sensors-24-06736-t004:** Normalized Signal Pulse Amplitude (NSPA) for the insect–material pairings of the stored-product recordings available in the BugBytes dataset, normalized using the median normalization method in the respective frequency bands.

Subject	Stage	Material	Sensor	Band [Hz]	NSPA [dB]
*P. interpunctella*	Larvae	Dog Food	Accelerometer	2000–3000	32.97
*P. interpunctella*	Larva	Dog Food	Piezoelectric	400–1200	30.79
*S. oryzae*	Larvae	Wheat Groats	PVDF film	4000–11,000	38.15
*S. oryzae*	Larvae	Wheat Groats	Accelerometer	2500–10,000	42.57
*S. oryzae*	Larvae	Wheat Groats	30 kHz Ultrasonic	2000–6000	11.74
*S. oryzae*	Larvae	Wheat Groats	40 kHz Ultrasonic	6000–9000	24.70
*S. oryzae*	Larvae	Wheat Groats	Piezoelectric	500–8000	37.49
*S. zeamais*	Larvae	Maize	Microphone	1000–4000	22.35
*S. zeamais*	Adults	Maize	Microphone	1000–6000	38.21
*P. trancatus*	Larvae	Maize	Microphone	1000–8000	36.72
*P. trancatus*	Adults	Maize	Microphone	1000–8000	29.67

## Data Availability

The data featured in this paper is publicly available on Kaggle as the A-SPIDS Stored Product Insect Dataset [80] and can be accessed using the Stored Product Insect Database available on GitHub [81].

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
