# Peer review of "Vibro-Acoustic Signatures of Various Insects in Stored Products"

_sensors, 2024, doi:10.3390/s24206736_

Round 1

Reviewer 1 Report

Comments and Suggestions for Authors

The manuscript has a fairly comprehensive overview of the current state of the art. The actual part describes the design and initial testing of the proposed test box and the method of signal processing and evaluation. Not infrequently, the experimental setup is disproportionately described without a detailed presentation of all the results (e.g. SNR testing of varying levels of natural and artificial noise).  The processing method is not described in detail using equations. The results are not treated in detail statistically, but the reasons are given in the paper (The insect material pairings are not fully balanced, also due to limited time and budget projection (lines 533-534)). Detailed description and analysis will probably be the content of future papers (lines 698-699).  Part Results should be better divided into more subclauses, especially the parts with own recordings and processing of records from datasets.   

160 resistor, two-resistor

166 dot at the end of the line is missing

541 The strength of the signal should be defined using some quantity.  

595 S_c, index c is missing

Author Response

Summary

Thank you very much for taking the time to review this manuscript. Please find the detailed responses below and the corresponding revisions and corrections changed in the re-submitted files.

Point-by-point response to Comments and Suggestions for Authors

Comment 1: The manuscript has a fairly comprehensive overview of the current state of the art. The actual part describes the design and initial testing of the proposed test box and the method of signal processing and evaluation. Not infrequently, the experimental setup is disproportionately described without a detailed presentation of all the results (e.g. SNR testing of varying levels of natural and artificial noise).

Response 1: Thank you for this comment. We want to describe the detection algorithm as well as the influence of natural and artificial noise in future publications. We tried to include initial results from the external noise influence on the considered metrics to the current paper but did not manage to do it within the 10 days provided for the paper improvement.

Comment 2: The processing method is not described in detail using equations.

Response 2: We added equations to describe the processing methods.

Comment 3: The results are not treated in detail statistically, but the reasons are given in the paper (The insect material pairings are not fully balanced, also due to limited time and budget projection (lines 533-534)). Detailed description and analysis will probably be the content of future papers (lines 698-699).

Response 3: Unfortunately, as noted by the reviewer, we do not have a fully balanced dataset. We hope that this publication will bring exposure to the topic and sensors, enabling more research and analysis on the matter. The data we did collect is publicly available and we will continue to add to the dataset as more recordings are made.

Comment 4: Part Results should be better divided into more subclauses, especially the parts with own recordings and processing of records from datasets.   

Response 4: Thank you for this suggestion. We created more subsections to better organize the paper.

Reviewer 2 Report

Comments and Suggestions for Authors

1. The introduction is too long. The author should summarize the status of the acoustic technology studied in this paper.

2. There are too many single Figures in the paper, so it is suggested that the author regroup the Figures.

3. It is suggested that the detection data or key indicators of the sensor be quantitatively described in the abstract.

Comments on the Quality of English Language

No.

Author Response

Summary

Thank you very much for taking the time to review this manuscript. Please find the detailed responses below and the corresponding revisions and corrections changed in the re-submitted files.

Point-by-point response to Comments and Suggestions for Authors

Comment 1: The introduction is too long. The author should summarize the status of the acoustic technology studied in this paper.

Response 1: Although we wanted to provide an extensive literature review, we do agree that it was quite lengthy. We removed collectively two pages from this section (minimizing it from 380 to 273 lines).

Comment 2: There are too many single Figures in the paper, so it is suggested that the author regroup the Figures.

Response 2: The paper has been updated with grouped figures.

Comment 3: It is suggested that the detection data or key indicators of the sensor be quantitatively described in the abstract.

Response 3: The abstract now includes more information about the NSEL and NSPA metrics.

Reviewer 3 Report

Comments and Suggestions for Authors

The paper contains the results of experiments carried out in the lab. The levels of noise emitted by four kinds of insects were measured by means of microphones and in the containers filled with edible substances by piezoelectric discs and microphones. 20 insect-material pairings were studied, noise spectra were analyzed and quotative metrics for evaluation of emitted noise were proposed.

The manuscript is suitable for the journal Sensors, it is clear enough and has important and interesting experimental results.

I recommend publishing the manuscript. However, I would like to ask the authors to consider some minor remarks:

1.       The frequency axis in all figure would be more informative in the logarithmic scale. The linear frequency scales do not show the low-frequency part of the spectrum.

2.       What numbers are shown in the brackets in Table 2?

3.       It is very important to provide information about the density of the insects (for example, the number of the insects or their larvae per liter) and to compare it with their density under real conditions.

4.       I believe that one decimal place is sufficient for all dB values in the manuscript.

Author Response

Response to Reviewer 3 Comments

Summary

Thank you very much for taking the time to review this manuscript. Please find the detailed responses below and the corresponding revisions and corrections changed in the re-submitted files.

Point-by-point response to Comments and Suggestions for Authors

Comment 1: The frequency axis in all figures would be more informative in the logarithmic scale. The linear frequency scales do not show the low-frequency part of the spectrum.

Response 1: Thank you for that suggestion. We chose to use the linear scale in accordance with other previous publications on the matter. Furthermore, the logarithmic scale emphasizes low frequency bands, which in our case is the range with most of the self-noise onto which we apply a bandpass filter to suppress the frequencies below 500 Hz.

Comment 2: What numbers are shown in the brackets in Table 2?

Response 2: The numbers in the parenthesis of Table 2 are supposed to detail the duration of external noise recordings for each respective insect and material pairing. Although we tried to indicate this in the caption, we want to thank the reviewer for confirming that it might not be clear. We decided to annotate the description directly into the cells of Table 2. We also separated the values of Table 3 into separate tables. 

Comment 3:  It is very important to provide information about the density of the insects (for example, the number of the insects or their larvae per liter) and to compare it with their density under real conditions.

Response 3: Thank you for raising awareness to this. We changed the units of the density from kg/m3 to kg/L. We also updated the description of the testing procedure (lines 412-423) to include comparison of our usage of 1 insect/liter to ISO/USDA standards and the other publications.

Comment 4:  I believe that one decimal place is sufficient for all dB values in the manuscript.

Response 4: We updated all the dB values in the manuscript to one decimal place. 

Reviewer 4 Report

Comments and Suggestions for Authors

This manuscript investigates the A-SPIDS system for detecting pests in stored products using acoustic detection algorithm based on signal amplitudes and energy levels. The structure of the paper is complete, the experimental design is sound, and the result analysis is clear. Before considering acceptance of this manuscript, the following issues should be addressed:

1. The paper only tests a few types of insects and materials. It is recommended to expand the range of tests to verify the algorithm's generalizability and robustness.

2. Only a simple noise suppression test is conducted. A more in-depth analysis is suggested, such as examining the effect of different noise types on detection performance and providing details on noise suppression algorithm design.

3. The paper uses only the signal-to-noise ratio as the evaluation metric. It is recommended to include other metrics, such as detection rate, false alarm rate, and missed detection rate, to provide a more comprehensive evaluation of the algorithm's performance.

4. The paper does not compare the proposed method with traditional signal processing methods. Does Hilbert transform necessary for feature extraction step?

Overall, the paper presents a promising approach for detecting insect infestations in stored products using acoustic signals. Addressing the weaknesses and implementing the recommended improvements would strengthen the paper and enhance its contribution to the field.

Comments on the Quality of English Language

/

Author Response

Summary

Thank you very much for taking the time to review this manuscript. Please find the detailed responses below and the corresponding revisions and corrections changed in the re-submitted files.

Point-by-point response to Comments and Suggestions for Authors

Comment 1: The paper only tests a few types of insects and materials. It is recommended to expand the range of tests to verify the algorithm's generalizability and robustness.

Response 1: Unfortunately, we do not have a fully balanced dataset because of limited time and budget as well as live insect access. We hope that this publication will bring exposure to the topic and sensors, enabling more research and analysis on the matter. This would include testing other types of insects and materials. We also hope to deploy the sensor within realistic settings such as inspection points and port facilities to examine its usage and external noise.

Comment 2: Only a simple noise suppression test is conducted. A more in-depth analysis is suggested, such as examining the effect of different noise types on detection performance and providing details on noise suppression algorithm design.

Response 2: Thank you for this comment. We want to have a more robust analysis on the detection and noise suppression algorithms in a future publication.  We tried to include initial results from the external noise influence on the considered metrics to the current paper but did not manage to do it within the 10 days provided for the paper improvement.

Comment 3: The paper uses only the signal-to-noise ratio as the evaluation metric. It is recommended to include other metrics, such as detection rate, false alarm rate, and missed detection rate, to provide a more comprehensive evaluation of the algorithm's performance.

Response 3: Thank you for this suggestion. We plan to include this information in a future publication when discussing the development of detection and noise suppression algorithms.

Comment 4: The paper does not compare the proposed method with traditional signal processing methods. Does Hilbert transform necessary for feature extraction step?

Response 4: We will discuss other signal processing methods in future publications. Since the insect signals materialize as impulses, the Hilbert transform enabled the measurement of the amplitude of these signals.

Round 2

Reviewer 2 Report

Comments and Suggestions for Authors

Recommended publication.

Comments on the Quality of English Language

no